# Averaged Optimization and Finite-Time Thermodynamics

**DOI:** 10.3390/e22090912

**Published:** 2020-08-20

**Authors:** Anatoly Tsirlin, Ivan Sukin

**Affiliations:** Ailamazyan Program Systems Institute of Russian Academy of Sciences, Petra Pervogo st., 4a, Veskovo, Yaroslavl oblast 152021, Russia

**Keywords:** averaged, optimization, thermodynamics, heat transfer, cyclic mode

## Abstract

The paper considers typical extremum problems that contain mean values of control variables or some functions of these variables. Relationships between such problems and cyclic modes of dynamical systems are explained and optimality conditions for these modes are found. The paper shows how these problems are linked to the field of finite-time thermodynamics.

## 1. Introduction

Averaging plays a very important role in optimization problems applied to engineering. Here are a few examples:1.Assume that there is a bound on the source flow $g_{s}$ and this constraint lowers the possible optimal value of the production. If we introduce some buffer (container) in such a way that the source flow is its feed, we can raise the possible value of the feed flow to the actual process without violating this constraint. Actual values of the feed flow will oscillate between values that are greater than and less than $g_{s}$. Only the mean value of the feed flow will be bounded in this case. Using this approach we can replace the strict constraint on $g_{s}$ by the averaged one. If we use such a buffer to store the product flow, we can maximize not this flow itself, but its mean value (Figure 1).Let us assume that the relationship between production rate *g* and consumption *q* has the form presented at Figure 2.With the help of buffers this relationship could be improved on interval from 0 to $q_{1}$. The process must operate with consumption $q_{1}$ during some fraction of time and with zero consumption during the remaining time. Relationship between average production and average consumption is represented by dashed slope line at Figure 2. This is the way pumps of water towers operate.2.Controls often can have only discrete values. For example, the light switch can be either on or off. None of these discrete values satisfy the constraints of the original problem. If there are devices that smooth out any oscillations of control variables, the optimal mode can correspond to the switching strategy that maintains given average values of flows. This kind of switching is the basis of electronic light dampers.3.In a heat engine the working fluid periodically makes contacts with the hot and cold sources, and the properties of these contacts must be chosen such that the average properties of the working fluid satisfy the constraints of the optimum cycle problem.

Averaged problems arise in finite-time thermodynamics for two main reasons:1.Many processes are periodic and their constraints must be satisfied on average per cycle.2.Interactions of thermodynamic systems are characterized by values of extensive variables *X* (volume, amount of substance, internal energy, entropy), and flows of mass and energy emerging in these interactions depend on intensive variables *y* (temperature, pressure, molar fraction). The rate of change of extensive variables depend on a flow, and of course on *y*. This means that the governing equations for thermodynamic interactions have the form:
(1)$$\frac{dX}{dt}=F\left(y\right)$$The right hand side of (Equation 1) does not contain *X* and this means that the increase in extensive variables during some given amount of time depends only on the mean value of *F*. It does not depend on the order in which intensive variables have different values, if the mean value of *F* remains constant. Equations such as (Equation 1) are called Lyapunov-type. They allow us to formulate the problem of optimal control for thermodynamic systems in averaged form.

Below we will consider some of these problems. The last section contains applications of methods developed in the paper to finite-time thermodynamics.

We consider dynamical systems characterized by a finite number of variables.

By a *steady-state mode* of a system we mean a mode such that, for every variable $y_{\nu }\left(t\right)$ characterizing the system, there exists a period $T_{\nu }$ such that the average value of $y_{\nu }\left(t\right)$ over this period is constant in time. Formally, this can be written as [1]
(2)$$\frac{1}{T_{\nu }}\int_{t-T_{\nu }}^{t}y_{\nu }\left(\tau \right)d\tau ={\bar{y}}_{\nu }.$$
Clearly, static modes, under which $y_{\nu }\left(t\right)$ are constant for all ν, satisfy this definition.

Another, more general, subclass of steady-state modes is formed by modes for which there exists a period *T* that is a multiple of all periods $T_{\nu }$. Such modes are called *cyclic* ones.

There are also steady modes for which there does not exist a common period *T* for all variables $y_{\nu }\left(t\right)$. This corresponds to the case where the ratio of at least two periods $T_{\nu }$ and $T_{\mu }$ is irrational. Such modes are called *quasi-cyclic* steady-state modes.

If the system is affected by external factors represented by stationary random processes and the mean values of the variables characterizing the system tend to some limits as the period *T* of averaging increases, then the steady mode is said to be *stochastic*.

A switch to a non-static steady-state mode may be caused either by the absence of a static mode admitted by the operating conditions of the system or by the fact that the efficiency of the system in a static mode is lower than in other modes.

Consider some examples.

The human organism in a steady mode in the absence of external perturbations is characterized by a constant temperature, a constant composition of arterial blood, etc. However, some factors such as the blood pressure and the lung volume periodically change. This is related to the ”structure“ of the respiration and circulation organs.A system consisting of a pump connected with a tank (e.g., a water tower) and consumers operates so that, even if the liquid consumption $\bar{G}$ is constant, the pump is sometimes completely switched off (and the liquid does not flow into the tank) and is sometimes switched on and operates with delivery higher than $\bar{G}$, with the average delivery being $\bar{G}$. If the dependence of the pump delivery *g* on the power expenditure *S* is described by a strictly convex function, then the average pump delivery is higher than that in the static mode at the same average power expenditure.In the rest of the paper, we mainly consider cyclic steady-state modes, among which two limit classes are distinguished. The first class includes modes in which each of the periods $T_{\nu }$ significantly exceeds the time of relaxation processes in the system. Moreover, each of the static steady states is assumed to be stable. In this case, we can neglect the dynamics of the system and assume that under variation of the mode variables, the state variables change in accordance with the static characteristics. Such modes are said to be *quasi-static*.The second class is formed by *sliding* steady-state modes, in which all or some of the control variables vary with frequency so high that, due to the inertia of the object, the state variables remain virtually constant, and their values depend only on the averaged influence of the control variables.Although static modes are a special case of cyclic modes, below by cyclic modes we mean modes under which at least one variable of the process changes periodically in time. A cyclic mode is said to be efficient if the passage to this mode improves the efficiency of the process in comparison with the static mode.Cyclic modes are typical of systems with no admissible static modes. Often a system has no static modes if the set *V* of admissible values of variables is non-convex; e.g., this set may include only discrete values. This is so, for example, in a heat engine in which a working fluid contacts a heat source whose temperature can take only two values, $T_{+}$ (a hot source) and $T_{-}$ (a cold source), and the average power over a cycle is required to be maximal under certain constraints.

Cyclic processes may be organized not only in time; variables may also depend on a spatial coordinate. In this case, the parameters of the system are constant in each section of the apparatus and vary periodically from section to section.

When passing from a static mode to a cyclic mode, one needs to replace the objective function by its mean value over the cycle and to replace all or some constraints imposed at each moment of time by averaged constraints. Thus, this passage involves an operation of averaging. Before passing to a cyclic mode, one must answer the following questions:Does there exist a cyclic mode satisfying the constraints of the problem?Is the transition from the optimal static mode to the cyclic mode efficient?What is the gain in the optimality criterion from this passage?What are the optimal forms of variation of the control and state variables, optimality conditions, computational algorithms?

It is desirable to answer questions 1–3 without solving problem 4, which is rather difficult in most cases.

Usually the problem of choosing an optimal static mode of an apparatus reduces to the problem of finding the extremal value of the objective function under certain equality and inequality constraints on the variables, i.e., to a non-linear programming problem. The transition to cyclic modes extends the set of possible solutions and, depending on a particular setting, leads either to an averaged non-linear programming problem or to a variational control problem. If the problem has an optimal static mode, then we refer to this problem as the initial problem.

In the rest of the paper we consider various methods for constructing problems with larger sets of admissible solutions as compared with the initial problem; we show that there are relationships between such extended problems, which allows one to estimate solutions and values of some of them by solving others.

## 2. Averaged Optimization Problems and Their Optimality Conditions

In this section, we consider various methods for introducing averaging into a non-linear programming (NLP) problem and obtain optimality conditions for averaged problems. To obtain these conditions, we use a trick based on reducing any averaged problem to a canonical form and deriving necessary optimality conditions for a particular problem from those for a general problem.

### 2.1. Averaging of Functions Included in the Formulation of an Optimization Problem

Consider an initial NLP problem [2] in the form
(3)$$f_{0}\left(x\right)\to \max\;/f_{i}\left(x\right)=0,\;i=\bar{1,m},\;x\in V_{x}.$$
On the set $V_{x}$, we define a probability measure p(x) such that
(4)$$\int_{V_{x}}p\left(x\right)dx=1,\;p\left(x\right)\geq 0.$$
The average value of the function f(x) on the interval [0,τ] can be calculated as follows:(5)$$\bar{f(x)}=\frac{1}{\tau }\int_{0}^{\tau }f\left(x\left(t\right)\right)dt=\int_{V_{x}}f\left(x\right)p\left(x\right)dx.$$
Let us assume that *x* varies with time or one solves the problem (Equation 3) and maximizes the mean value of $f_{0}$, but not the value of this function itself. If functions $f_{i}$ vanish on average, then we will arrive at a problem of the form
(6)$$\bar{f_{0}\left(x\right)}\to \max\;/\bar{f_{i}\left(x\right)}=0,\;i=\bar{1,m}.$$
A sought solution of problem (Equation 6) is a measure p(x) on $V_{x}$ rather than a vector *x*. The variable *x* is called a *randomized* one, and p(x) is called a *generalized* solution. Following A.D. Ioffe and V.M. Tikhomirov [3], we call the value of the objective functional at the optimal solution as *the value of a problem*.

### 2.2. Convex Hulls—Carathéodory’s Theorem

The notion of *convexity* is very important for optimization problems.

1The convex hull of a set *V* is the minimum convex set CoV such that V⊂CoV.2The set of points lying on or below the graph of a function is called its *hypograph*. The convex hull Cof of a function *f* is the upper boundary of the convex hull of its hypograph.3Alternatively, the convex hull of a function *f* is the minimum convex function defined on the convex hull of the domain of *f*. For every $\tilde{x}$ from the domain of *f* the following holds: $Cof\left(\tilde{x}\right)\geq f\left(\tilde{x}\right)$.

Carathéodory’s theorem is the most important theorem of convex analysis and geometry. It states that coordinates of every point of the convex hull of the set $V\subset R^{n}$ could be calculated as the weighted arithmetic mean of some points of *V* and the maximum necessary number of these points is no more than n+1. The beautiful exposition of this theorem is given in [4].

### 2.3. Optimal Distribution in An Averaged NLP Problem

Let us take some $x_{0}\in V_{x}$. If $p\left(x\right)=\delta \left(x-x_{0}\right)$, then problem (Equation 6) coincides with the initial problem. If the set of admissible solutions of a problem includes the set of admissible solution of the initial NLP problem and the optimality criteria in both problems coincide on the set of admissible solutions of the NLP problem, then the former problem is called an extension of the NLP problem.

First, consider the special form of problem (Equation 6) with $f_{i}\left(x\right)=x_{i}$:(7)$$\bar{f_{0}\left(x\right)}\to \max\;/{\bar{x}}_{i}=0,\;i=\bar{1,n}.$$
The value of the problem (Equation 7) is equal to the ordinate of the convex hull of the function $f_{0}\left(x\right)$ on the set $V_{x}$ at the point x=0. According to Carathéodory’s theorem, constructing any ordinate of the convex hull of a function of n variables requires averaging at most n+1 ordinates of the function $f_{0}\left(x\right)$; therefore, we can rewrite problem (Equation 7) in the form
(8)$$\sum_{\nu =0}^{n}\gamma_{\nu }f_{0}\left(x^{\nu }\right)\to \max\;/\sum_{\nu =0}\gamma_{\nu }x_{i}^{\nu }=0,\;i=\bar{1,n},\;\gamma_{\nu }\geq 0,\;\sum_{\nu =0}^{n}\gamma_{\nu }=1.$$
Let us return to problem (Equation 6) and try to reduce it to simple calculation. We need to calculate the ordinate of a convex hull of the given function. Please note that problem (Equation 6) can be solved in two stages. At the first stage, we find the maximum of the function $f_{0}\left(x\right)$ subject to the constraint f(x)=C, where *C* takes all values for which the level surface f(x)=C intersects $V_{x}$. The problem
(9)$$f_{0}\left(x\right)\to \max/f_{i}\left(x\right)=C,\;i=\bar{1,m},\;x\in V_{x}$$
is a non-linear programming problem. Solving (Equation 9), we obtain a set of conditionally optimal solutions $x^{*}\left(C\right)$ and the corresponding values of the *reachability function*
$f_{0}^{*}\left(C\right)=f_{0}\left(x^{*}\left(C\right)\right)$ of the non-linear programming problem.

The following assertion holds: *The optimal distribution $p^{*}\left(x\right)$ in problem (Equation 6) is concentrated at the points $x^{*}\left(C\right)$*. In other words, *one needs to average only over conditionally optimal values of $f_{0}$*.

### 2.4. Necessary Conditions of Optimality—Kuhn-Tucker Theorem

The Kuhn-Tucker theorem generalizes Lagrange multipliers method to problems with inequality constraints:(10)$$f_{0}\left(x\right)\to \max_{x}\;/f_{i}\left(x\right)=0,\;\varphi_{j}\left(x\right)\geq 0,\;i=\bar{1,k},\;j=\bar{k+1,m},$$
where all functions are smooth.

The theorem states that *there is nonzero vector of Lagrange multipliers with components $\lambda_{i}$, $\mu_{j}\leq 0$ such that Lagrange function*
(11)$$L=\lambda_{0}f_{0}\left(x\right)+\sum_{i}\lambda_{i}f_{i}\left(x\right)+\sum_{j}\mu_{j}\varphi_{j}\left(x\right)=R\left(\lambda ,x\right)+\sum_{j}\mu_{j}\varphi_{j}\left(x\right)$$
*is stationary on the optimal solution of the problem (Equation 10). The multiplier $\lambda_{0}$ could equal to zero or one. In the former case the solution is called degenerate.*


It follows from this theorem that when $\varphi_{j}\left(x\right)=x_{j}$ we have inequality $\frac{\partial R}{\partial x_{j}}\leq 0$ for the optimal solution. More detailed explanation could be found in [5].

### 2.5. Reduction to an Ordinary NLP Problem

The above considerations allow us to formulate the second stage in solving problem (Equation 6). This stage is the maximization of the average value of the function $f_{0}^{*}\left(C\right)$ with the constraint that the vector *C* has zero mean, i.e.,
(12)$$\bar{f_{0}^{*}\left(C\right)}\to \max\;/{\bar{C}}_{i}=0,\;i=\bar{1,m},\;C_{i}\in V_{C}.$$
This problem is similar to the problem (Equation 7). Its value, and hence the value of problem (Equation 6), is equal to the ordinate of the convex hull of the reachability function $f_{0}^{*}\left(C\right)$ at C=0:(13)$$\underset{x\in D}{\sup}\bar{f_{0}\left(x\right)}=\sup\bar{f_{0}^{*}\left(C\right)}\;/\bar{C}=0,\;C\in V_{C}.$$
Since the vector *C* is *m*-dimensional, the number of base points $C^{\nu }$ in problem (Equation 9) is at most m+1. Thus, the distribution p(C) in problem (Equation 9) can be sought in the form
(14)$$p\left(C\right)=\sum_{\nu =0}^{m}\gamma_{\nu }\delta \left(C-C^{nu}\right).$$
Since each of the base values $C^{\nu }$ corresponds to a conditionally optimal solution $x^{*}\left(C^{\nu }\right)$, the optimal distribution p(x) is also concentrated at no more than m+1 points:(15)$$p\left(x\right)=\sum_{\nu =0}^{m}\gamma_{\nu }\delta \left(x-x^{\nu }\right).$$
Substituting the distribution (Equation 15) into the expressions for $\bar{f_{0}\left(x\right)}$ and $\bar{f_{i}\left(x\right)}$, we reduce problem (Equation 6) to the form
(16)$$\begin{array}{c} I=\sum_{\nu =0}^{m}\gamma_{\nu }f_{0}\left(x^{\nu }\right)\to \max\;/\sum_{\nu =0}^{m}\gamma_{\nu }f_{i}\left(x^{\nu }\right)=0,\\ i=\bar{1,m},\;x^{\nu }\in V_{x},\;\gamma_{\nu }\geq 0,\;\sum_{\nu =0}^{m}\gamma_{nu}=1. \end{array}$$
Thus, we have reduced the problem to an ordinary NLP problem whose variables are the base values $x^{\nu }$ of the vector *x* and the weight factors $\gamma_{\nu }$.

### 2.6. Relationship between Averaged NLP Problem and the Lagrangian Function of the NLP Problem without Averaging

The Lagrangian function
(17)$$\begin{array}{c} \bar{R}=\sum_{\nu =0}^{m}\gamma_{\nu }f_{0}\left(x^{\nu }\right)+\sum_{i=1}^{m}\lambda_{i}\sum_{\nu =0}^{m}\gamma_{\nu }f_{i}\left(x^{\nu }\right)+\Lambda \left(1-\sum_{\nu =0}^{m}\gamma_{\nu }\right)=\\ =\sum_{\nu =0}^{m}\gamma_{\nu }\left[f_{0}\left(x^{\nu }\right)+\sum_{i=1}^{m}\lambda_{i}f_{i}\left(x^{\nu }\right)-\Lambda \right]+\Lambda \end{array}$$
of problem (Equation 16) is related to the Lagrangian function
(18)$$R=f_{0}\left(x\right)+\sum_{i=1}^{m}\lambda_{i}f_{i}\left(x\right)$$
of the initial NLP problem by
(19)$$\bar{R}=\sum_{\nu =0}^{m}\gamma_{\nu }\left(R\left(x^{\nu },\lambda \right)-\Lambda \right)+\Lambda .$$
Since $\sum_{\nu }\gamma_{\nu }=1$, the Lagrangian function of the averaged problem equals the average value of the Lagrangian function of the initial problem over all base values $x^{\nu }$. Some of the weight factors $\gamma_{\nu }$ may vanish; then the number of base points is less than m+1.

Let us find conditions that must hold for those $x^{\nu }$ that have non-zero weights in (Equation 19). For this purpose, we apply the Kuhn–Tucker theorem and write the optimality conditions for problem (Equation 16) with respect to the variables $\gamma_{\nu }$:(20)$$\frac{\partial \bar{R}}{\partial \gamma_{\nu }}\delta \gamma_{\nu }\leq 0.$$
Since $\gamma_{\nu }$ are bounded only from below ($\gamma_{\nu }\geq 0$), it follows that $\delta \gamma_{\nu }\geq 0$; therefore,
(21)$$\frac{\partial \bar{R}}{\partial \gamma_{\nu }}=R\left(x^{\nu }\right)-\Lambda \leq 0,$$
or $R(x^{\nu })\geq \Lambda$. If $\gamma_{\nu }^{*}>0$, then $\delta \gamma_{\nu }$ may be of any sign, and so inequality (Equation 21) transforms into the equality
(22)$$R(x^{\nu })=\Lambda .$$
Thus, *for all $x^{\nu }$ involved in the averaged problem with non-zero weights, the Lagrangian function R of the initial non-linear programming problem attains an absolute maximum*. Of course, this maximum is the same for all $x^{\nu }$.

The requirements that the function *R* must take the same value at all points $x^{\nu *}$ and that this value must be maximum give equations for the variables to be found. Thus, applying Kuhn–Tucker theorem the problem (Equation 6), we obtain the vector of Lagrange multipliers λ for which the function $\bar{R}$ attains an absolute maximum with respect to the variables $x^{\nu }\in V_{x}$ and $\gamma_{\nu }\in V_{\gamma }$ at an element of the set *D* of admissible solutions to problem (Equation 6), and these multipliers λ satisfy the condition
(23)$$\bar{R}\left(\lambda^{*},\gamma_{\nu }^{*},x^{\nu *}\right)=\underset{\lambda \in V_{\lambda }}{\inf}\underset{\gamma_{\nu },x^{\nu }}{\sup}\bar{R}\left(\lambda ,\gamma_{\nu },x^{\nu }\right)=\underset{\lambda \in V_{\lambda }}{\inf}\underset{x\in V_{x}}{\sup}R\left(\lambda ,x\right).$$
Thus, when the attainability function $f_{0}^{*}\left(C\right)$ coincides with its convex hull at C=0, the transition to the averaged problem is not efficient (the values of the NLP problem and problem (Equation 6) coincide). By virtue of (Equation 23), we can look for the value of the averaged problem in the form $\underset{\lambda \in V_{\lambda }}{\inf}\underset{x\in V_{x}}{\sup}R\left(\lambda ,x\right)$. If the extended problem is inefficient, then we say that it is equivalent to the initial problem.

In the general case, the dimension of the vector of unknown variables and the computational complexity of problem (Equation 6) are much greater than those for the NLP problem. However, in many cases, we are interested not in the solution but in the value of the averaged problem, which shows the gain obtained by transition to the averaged setting. Some methods for estimating the value of problem (Equation 6) from above and below were proposed in [6].

### 2.7. Other Forms of Averaged Extensions of the NLP Problem

Problem (Equation 6) is not the only possible extension of the NLP problem by averaging. The optimality criteria, relations, and constraints in real-life problems often include the mean values of variables *x* rather than the variables themselves. For example, the performance of a distillation column is characterized by the mean not current composition of output flows, because these flows are accumulated in some containers or apparatuses at the exit of the column (or attached to the column). Below, we describe several possible modifications of the averaged extension [7].

*Problem of maximizing a function of the mean value of the argument.* When *D* is the set of admissible solutions of the initial NLP problem, i.e., *D* is defined by the condition f(x)=0, and $\bar{x}$ is the mean value of the vector *x* on the set *D*, we have:
(24)$$f_{0}\left(\bar{x}\right)\to \sup\;/p\left(x\right)=0\;\forall x\notin D.$$Since the set of values $\bar{x}$ satisfying this condition is the convex hull of *D*, problem (Equation 24) is equivalent to the NLP problem on the convex hull of *D*:
(25)$$f_{0}\left(x\right)\to \sup\;/x\in \mathrm{Co}D.$$*Problem of maximizing the mean value of a function under constraints imposed on the mean value of the argument:*(26)$$\bar{f_{0}\left(x\right)}\to \sup\;/f_{i}\left(\bar{x}\right)=0,\;i=\bar{1,m}$$
or, in more detail,
(27)$$\int_{V_{x}}f_{0}\left(x\right)p\left(x\right)dx\to \underset{p(x)}{\sup}\;/f_{i}\left(\int_{V_{x}}x\cdot p\left(x\right)dx\right)=0,\;i=\bar{1,m}.$$
*Problem of maximizing a function of the mean value of x under averaged constraints:*
(28)$$f_{0}\left(\bar{x}\right)\to \sup\;/\bar{f_{i}\left(x\right)}=0,\;i=\bar{1,m}.$$


Each of the above problems is an extension of the non-linear programming problem, and the solutions of these problems are distributions p(x).

*Averaged problems with two types of variables*. An NLP problem can be extended only with respect to some components of the solution rather than with respect to the whole solution. In practice, this situation occurs when the problem is solved repeatedly and some components (we denote them by *x*) can vary from one solution to another, while the remaining components must be chosen only once and then fixed. We denote the latter group of variables by *y*. For example, *x* may be the operating conditions of the process (such as flow, pressure, temperature, etc.) and *y* may be the design parameters of an apparatus.

If we denote
(29)$${\bar{f(y,x)}}^{x}=\int_{C_{x}}f\left(y,x\right)p\left(x\right)dx,$$
a problem in which averaging is performed over only part of variables has the form
(30)$${\bar{f_{0}\left(y,x\right)}}^{x}\to \sup\;/{\bar{f_{i}\left(y,x\right)}}^{x}=0,\;i=\bar{1,m}.$$
One need to find the vector *y* and distribution p(x) in (Equation 30).

For each fixed *y*, this problem coincides with the usual setting of problem (Equation 6). If we separate the randomized variables $x\in E^{r}$ and the deterministic variables $y\in E^{s}$ in the Lagrangian function *R* of the initial NLP problem, then we can write optimality conditions with respect to *x* by analogy with problem (Equation 6) in the form (see (Equation 23))
(31)$$R\left(\lambda ,\gamma_{\nu }^{*},y,x^{\nu *}\right)=\underset{x\in V_{x}}{\sup}R\left(\lambda ,y,x\right),\;\nu =\bar{0,m}.$$
In this case, if we denote the admissible set of (Equation 30) as ${\bar{D_{x}\left(y\right)}}^{x}$, for each $y\in V_{y}$, there exist λ(y) such that
(32)$$\underset{\lambda }{\inf}\underset{x\in V_{x}}{\sup}R\left(\lambda ,y,x\right)=\underset{x\in {\bar{D_{x}\left(y\right)}}^{x}}{\sup}\bar{f_{0}\left(y,x\right)}.$$
The Lagrangian function attains an absolute maximum at the base values of *x*.

At the same time, for a fixed function p(x), problem (Equation 30) becomes a usual non-linear programming problem with respect to the variables *y*. The Kuhn–Tucker conditions hold for this problem, which include in this case the complementary slackness conditions as well as the requirement that the function $R(\lambda ,\gamma_{\nu },y,x^{\nu })$ be stationary with respect to *y*, which in turn, leads to the equations
(33)$$\frac{\partial }{\partial y_{j}}\left[\sum_{\nu =0}^{m}\gamma_{\nu }R\left(\lambda ,y,x^{\nu }\right)\right]=0,\;j=\bar{1,s}.$$
where *R* is the Lagrangian function for the NP problem.

Averaged problems with two types of variables are in a sense close to optimal control problems, and optimality conditions for such problems are close to the Pontryagin maximum principle.

### 2.8. The Algorithm for Obtaining Optimality Conditions in Averaged Problems

By *an averaged problem of static optimization* we mean any NLP problem in which either functions or variables are averaged with respect to all or part of the variables.

As shown above, the settings of averaged problems are very diverse. The reason for this is that a problem may contain both the mean values of functions and functions of the mean values of variables. Moreover, averaging may involve only part of the variables. Under these conditions, it is inexpedient to study each possible setting of an averaged problem. It is significantly more convenient to obtain optimality conditions for some canonical form of an averaged problem and apply them to each particular problem after having reduced the latter to this canonical form [8].

Before obtaining optimality conditions, we must answer the following two questions:Is the optimal distribution, which is one of the components of the solution of an averaged problem, always concentrated at finitely many base points?If the answer to the previous question is ”yes,“ then what is the limit number of these points?

The necessary optimality conditions given below yield an affirmative answer to the first question and allow one to determine the limit number of base points.

Let *y* denote the vector of deterministic variables, and let *x* be the vector of randomized variables. For the former, we must find an optimal value, and for the latter, an optimal measure. The canonical form of the averaged problem is
(34)$$F_{0}\left(\bar{f(x,y)},y,x\right)\to \max$$
under the constraints
(35)$$\begin{array}{c} F_{\nu }\left(\bar{f(x,y)},\varphi \left(x,y\right),\bar{x}\right)=0,\;\nu =\bar{1,r},\\ F_{\nu }\left(\bar{f(x,y)},\varphi \left(x,y\right),\bar{x}\right)\geq 0,\;\nu =\bar{r+1,m}. \end{array}$$
Here the bar over the symbol of a function denotes averaging over the set $V_{x}$ of randomized variables *x*, which is assumed to be compact.

Suppose that the vector *x* has dimension *k* and the vector function *f* has dimension *n*. The function *F* is assumed to be continuously differentiable with respect to all its variables, and *f* and φ are continuous in *x* and continuously differentiable in *y*.

In [8], one of the authors (*A.T.*) proved that *the optimal measure $p^{*}\left(x\right)$ on the set of randomized variables is concentrated at no more than L+1 base points, where L=n+k*. Thus,
(36)$$p^{*}\left(x\right)=\sum_{l=0}^{L}\gamma_{l}\delta \left(x-x^{l}\right),\;\gamma_{l}\geq 0,\;\sum_{l=0}^{L}\gamma_{l}=1.$$
Therefore, for the optimal solution, we have
(37)$$\bar{f^{*}\left(x,y\right)}=\sum_{l=0}^{L}\gamma_{l}f\left(x^{l},y\right),\;\bar{x}=\sum_{l=0}^{L}\gamma_{l}x^{l},$$
and constraints (Equation 35) take the form
(38)$$\begin{array}{c} F_{\nu }\left(\bar{f},\varphi \left(x^{l},y\right),\bar{x}\right)=0,\;\nu =\bar{1,r},\\ F_{\nu }\left(\bar{f},\varphi \left(x^{l},y\right),\bar{x}\right)\geq 0,\;\nu =\bar{r+1,m}. \end{array}$$
for all values of $x^{l}$.

These expressions turn problem (Equation 34), (Equation 35) into an ordinary NLP problem with respect to $\gamma_{l}$, *y* and $x^{l}$. The Kuhn–Tucker conditions reduce to the following: the Lagrangian function
(39)$$R=F_{0}\left(\bar{f},y,\bar{x}\right)+\sum_{\nu =1}^{m}\lambda_{\nu }F_{\nu }\left(\bar{f},\varphi \left(x^{l},y\right),\bar{x}\right)$$
of this problem is stationary with respect to $x^{l}$ and *y* and is unimprovable with respect to $\gamma_{l}$ (we assume the solution is non-degenerate, so $\lambda_{0}=1$). To write down the optimality conditions, we introduce the notation
(40)$$a_{j}=\frac{\partial R}{\partial {\bar{f}}_{j}},\;\beta_{i}=\frac{\partial R}{\partial x_{i}},\;r_{\mu l}=\frac{\partial R}{\partial \varphi_{\mu }}\left(x^{l},y\right).$$
Using this notation, we can write the condition that *R* is unimprovable with respect to $\gamma_{l}$ as follows: the expression
(41)$$C\left(x\right)=\sum_{j}a_{j}f_{j}\left(x,y^{*}\right)+\sum_{i}\beta_{i}x_{i}$$
attains its maximum with respect to $x\in V_{x}$ at the points $x^{l}$, so that
(42)$$x^{l*}=\underset{x}{\arg\max}C\left(x\right),\;l=\bar{1,L};$$
the condition that *R* is stationary with respect to *y* has the form
(43)$$\nabla_{y}\left[\sum_{j}a_{j}\bar{f_{j}\left(x,y\right)}+F_{0}\left(\bar{f},x,y\right)+\sum_{\mu ,l}r_{\mu l}\varphi_{\mu }\left(x^{l},y\right)\right]=0.$$
The maximality of C(x), together with equations (Equation 42), constraints (Equation 35), and the complementary slackness conditions
(44)$$\sum_{\nu =r+1}^{m}\lambda_{\nu }F_{\nu }\left({\bar{f}}^{*},\varphi^{*},x^{*}\right)=0,\;\lambda_{\nu }\geq 0,\;\nu =\bar{r+1,m}$$
allows one to find a solution $\gamma_{l}^{*}$, $y^{*}$, $x^{l}$.

When formulating a specific averaged problem, one
writes the conditions of the problem in the canonical form (Equation 34), (Equation 35);separates the randomized and deterministic variables;calculates the total number *L* of averagings, which is equal to the sum of the dimensions of the vector of randomized variables and of the vector of functions to be averaged;constructs the functions *R* and *C* and substitutes them into expressions (Equation 42)–(Equation 44).

For example, in problem (Equation 26), we have
(45)$$F_{0}=\bar{f_{0}\left(x\right)},\;F_{\nu }=f_{\nu }\left(\bar{x}\right),\;\nu =\bar{1,m}.$$
The number *L* equals *k*, and
(46)$$R=\lambda_{0}\bar{f_{0}\left(x\right)}+\sum_{\nu =1}^{m}\lambda_{\nu }f_{\nu }\left(\bar{x}\right).$$
In (Equation 42), we have $a_{0}=\lambda_{0}=1$, $a_{\nu }=0$ for ν>0 and
(47)$$\beta_{i}=\sum_{\nu =1}^{m}\lambda_{\nu }{\left(\frac{\partial f(\bar{x})}{\partial {\bar{x}}_{i}}\right)}_{\bar{x}},\;i=\bar{1,k}.$$
At the base points $x^{l}$, the number of which does not exceed k+1, the expression
(48)$$C\left(x\right)=f_{0}\left(x\right)+\sum_{i=1}^{k}x_{i}\sum_{\nu =1}^{m}\lambda_{\nu }{\left(\frac{\partial f(\bar{x})}{\partial {\bar{x}}_{i}}\right)}_{x^{*}}$$
attains its maximum, and conditions (Equation 35) hold, which have the form
(49)$$f_{\nu }\left(\sum_{l=0}^{k}\gamma_{l}x^{l}\right)=0,\;\nu =\bar{1,m}.$$

## 3. Non-Stationary Problems of Averaged Optimization

Consider an extremal problem of the form
(50)$${\bar{f}}_{0}=\frac{1}{\tau }\int_{0}^{\tau }f_{0}\left(J\left(t\right),u\left(t\right)\right)dt\to \max_{u}$$
subject to the constraints
(51)$${\bar{f}}_{\nu }=\frac{1}{\tau }\int_{0}^{\tau }f_{\nu }\left(J\left(t\right),u\left(t\right)\right)dt=0,\;\nu =\bar{1,n},$$
where the functions $f_{\nu }:R^{k_{1}}\times R^{k_{2}}\to R$, $\nu =\bar{0,n}$, are continuous in *J* and *u*, $u\in V_{u}\subset R^{k_{1}}$ is a measurable function, the set $V_{u}$ is compact, and $J\left(t\right)\in V_{J}\subset R^{k_{2}}$ is a given measurable function of time. With J(t) we can associate a probability measure (distribution) p(J). If J(t) takes a value $J^{k}$ on a part of the interval (0,τ) of relative length $\alpha_{k}$, then p(J) contains a term of the form $\alpha_{k}\delta \left(J-J^{k}\right)$. The length of the interval (0,τ) may tend to infinity, and J(t) may be a stationary random process with distribution p(J).

The distribution p(J) can be written in the form
(52)$$p\left(J\right)=\bar{p}\left(J\right)+\sum_{k}\alpha_{k}\delta \left(J-J^{k}\right).$$
For problem (Equation 50), (Equation 51), let ατ be the length of the part of (0,τ) on which J(t) takes one of the constant values $J^{k}$; we have $\alpha =\sum_{k}\alpha_{k}$. We refer to ατ as the total constancy interval of J(t). The remaining part (1−α)τ is called the interval of variation of the parameter *J*.

**Theorem** **1.**
*Let $u^{*}\left(t\right)$ be an optimal solution; then there exists a non-zero vector $\lambda =\{\lambda_{0},\ldots ,\lambda_{n}\}$ with $\lambda_{0}\in \left\{0,1\right\}$ such that*

*on the interval of variation of the parameter J(t)*
(53)$$u^{*}\left(J,\lambda \right)=\underset{u\in V_{u}}{\arg\max}\sum_{\nu =0}^{n}\lambda_{\nu }f_{\nu }\left(J,u\right);$$

*on the total constancy interval of J(t), the optimal solution switches between at most n+1 base values $u^{j}$, and each of these values satisfies the condition*
(54)$$u^{j}=\underset{u\in V_{u}}{\arg\max}\sum_{k}\alpha_{k}\sum_{\nu =0}^{n}\lambda_{\nu }f_{\nu }\left(J^{k},u\right),\;j=\bar{0,n};$$

*the portions $\gamma_{j}$ of the constancy interval ατ on which $u^{*}\left(t\right)$ takes the respective values $u^{j}$ satisfy the conditions*
(55)$$\begin{array}{c} \int_{V_{J}}\bar{p}\left(J\right)f_{\nu }\left(J,u^{*}\left(J\right)\right)dJ+\sum_{j=0}^{n}\gamma_{j}\sum_{k}\alpha_{k}f_{\nu }\left(J^{k},u^{j}\right)=0,\;\nu =\bar{1,n},\\ \sum_{j=0}^{n}\gamma_{j}=1,\;\gamma_{j}\geq 0; \end{array}$$

*the vector of multipliers $\lambda_{\nu }$, $\nu =\bar{1,n}$, is determined by the conditions*
(56)$$\lambda^{*}=\arg\min_{\lambda }\left[\int_{V_{J}}\bar{p}\left(J\right)\sum_{\nu =0}^{n}\lambda_{\nu }f_{\nu }\left(J,u^{*}\left(J,\lambda \right)\right)dJ+\sum_{j=0}^{n}\gamma_{j}\sum_{\nu =0}^{n}\lambda_{\nu }\sum_{k}\alpha_{k}f_{\nu }\left(J^{k},u^{j}\left(\lambda \right)\right)\right].$$



Thus, on the constancy intervals, the optimal solution of a problem with non-stationary parameters coincides with the solution of an averaged mathematical programming problem, and on the interval of variation of the parameter, it varies as the solution of a problem with integral constraints. This theorem was proved in [9].

**Example** **1.**
*Consider the problem of maximizing the average power $\bar{p}$ of a heat engine in which the working fluid contacts a source of variable temperature $T_{0}\left(t\right)$. This problem has the form*
(57)$$\bar{p}=\frac{1}{\tau }\int_{0}^{\tau }q\left(T_{0}\left(t\right),T\left(t\right)\right)dt\to \max_{T}$$
*subject to the constraint*
(58)$$\bar{\sigma }=\frac{1}{\tau }\int_{0}^{\tau }\frac{q(T_{0}\left(t\right),T\left(t\right))}{T(t)}dt=0.$$
*Here T(t) is the temperature of the working substance, q is the heat flux from the source to the working fluid, and $\bar{\sigma }$ is the mean rate of variation of the entropy of the working substance. A substantiation of the setting (Equation 56), (Equation 57) can be found in [10,11,12]. The optimality conditions (Equation 53) imply the following relation for the interval of variation of $T_{0}\left(t\right)$:*
(59)$$\frac{1}{T^{2}}\frac{q(T_{0},T)}{\partial q(T_{0},T)/\partial T}-\frac{1}{T}=\mathrm{const}.$$


In particular, for the Newtonian law $q\left(T_{0},T\right)=\beta \left(T_{0}-T\right)$ of heating, (Equation 59) implies
(60)$$T^{*}\left(T_{0}\right)=m\sqrt{T_{0}},$$
where *m* is the constant equal to the mean value of the square root of the source temperature.

For example, suppose that $T_{0}\left(t\right)$ has a uniform distribution (for a regular function $T_{0}\left(t\right)$, this means that the source temperature depends linearly on time) and $T_{02}$ and $T_{01}$ are the maximal and minimal source temperatures, respectively. Then
(61)$$T^{*}\left(T_{0}\right)=\frac{2(T_{02}^{3/2}-T_{01}^{3/2})}{3(T_{02}-T_{01})}\sqrt{T_{0}}.$$
The maximum power is given by
(62)$${\bar{p}}_{max}=\beta \left[\frac{T_{02}+T_{01}}{2}-\frac{4}{9}\frac{\left(T_{02}^{3/2}-T_{01}^{3/2}\right)}{T_{02}-T_{01}}\right].$$
Thus, a heat engine with one source may have non-zero power if the variance of the source temperature is positive.

For some laws $q(T_{0},T)$, the optimal temperature $T^{*}\left(t\right)$ may switch between two base values on intervals of constancy of the parameter $T_{0}$.

## 4. Estimation of the Performance of Cyclic Modes

Suppose that the dynamics of a system is characterized by the differential equations
(63)$${\dot{x}}_{\nu }=f_{\nu }\left(x,u,a\right),\;\nu =\bar{1,m},$$
whose right-hand sides do not explicitly depend on *t*. Here, as in the preceding sections, *x* denotes the state variables, *u* are the control ones, and *a* denotes parameters to be optimized. As a rule, boundary conditions are not fixed for equations (Equation 63), but the state variables are required to vary periodically:(64)$$x_{\nu }\left(\tau \right)=x_{\nu }\left(0\right)\Rightarrow \int_{0}^{\tau }f_{\nu }\left(x,u,a\right)dt=0,\;\nu =\bar{1,m}.$$
The performance averaged over the cycle plays the role of the optimality criterion for such a cyclic process and can be written in the form
(65)$$I=\frac{1}{\tau }\int_{0}^{\tau }f_{0}\left(x,u,a\right)dt\to \max.$$
The duration τ of each cycle is one of the components of the vector *a*; in the general case, it is not fixed. The parameters and controls are subject to constraints $a\in V_{a}$ and $u\in V_{u}$; in addition to the integral constraints (Equation 64), which follow from the periodicity of the process, the problem usually contains integral constraints determined by given mean rates of consumption of some resources (resource constraints):(66)$$J_{j}=\int_{0}^{\tau }\varphi_{j}\left(x,u,a\right)dt=0,\;j\in \bar{1,r}.$$
It is assumed that each of the functions determining the problem is continuous in all its variables and is continuously differentiable with respect to *x* and *a*.

**Optimality conditions**. Optimality conditions for problem (Equation 63)–(Equation 66) can be obtained by using the maximum principle [6]. Namely, if an optimal solution $x^{*}$, $a^{*}$, $u^{*}$ exists and is non-degenerate, then there exist a non-zero vector λ and a differentiable vector function ψ(t) such that the function
(67)$$R=\frac{1}{\tau }f_{0}+\sum_{\nu }\left[{\dot{\psi }}_{\nu }x_{\nu }+\left(\psi_{\nu }+\lambda_{\nu }\right)f_{\nu }\right]+\sum_{j}\lambda_{j}\varphi_{j}.$$
is stationary with respect to *x* and attains a maximum with respect to *u*, and the integral *S* of this function is locally unimprovable with respect to *a*. Thus,
(68)$$\frac{\partial R}{\partial x_{i}}=0\;\Rightarrow \;{\dot{\psi }}_{i}=-\frac{\partial }{\partial x_{i}}\left\{\frac{1}{\tau }f_{0}+\sum_{\nu }\left(\psi_{\nu }+\lambda_{\nu }\right)f_{\nu }+\sum_{j}\lambda_{j}\varphi_{j}\right\}.$$
Since the values $x_{\nu }\left(\tau \right)$ and $x_{\nu }\left(0\right)$ are not fixed, it follows that $\psi_{\nu }\left(\tau \right)$ and $\psi_{\nu }\left(0\right)$ vanish. Introducing the notation ${\tilde{\psi }}_{\nu }=\psi_{\nu }+\lambda_{\nu }$ and taking into account the equality ${\dot{\tilde{\psi }}}_{\nu }={\dot{\psi }}_{\nu }$, we can rewrite condition (Equation 68) in the form
(69)$${\dot{\tilde{\psi }}}_{i}=-\frac{\partial }{\partial x_{i}}\left\{\frac{1}{\tau }f_{0}+\sum_{\nu }\psi_{\nu }f_{\nu }+\sum_{j}\lambda_{j}\varphi_{j}\right\}=-\frac{\partial }{\partial x_{i}}H.$$
For these equations, since ψ(0) and ψ(τ) vanish, the costate variables satisfy the periodicity conditions
(70)$${\tilde{\psi }}_{\nu }\left(0\right)={\tilde{\psi }}_{\nu }\left(\tau \right)\;\Rightarrow \;\int_{0}^{\tau }\frac{\partial H}{\partial x_{\nu }}dt=0,\;\nu =\bar{1,m}.$$
The conditions of maximality of *R* with respect to *u* have the form
(71)$$u^{*}\left(t\right)=\underset{u\in V_{u}}{\arg\max}\left\{\frac{f_{0}}{\tau }+\sum_{\nu }{\tilde{\psi }}_{\nu }f_{\nu }+\sum_{j}\lambda_{j}\varphi_{j}\right\}.$$
Finally, the optimality conditions with respect to each component $a_{k}$ of the vector *a*, including the duration τ of the cycle, yield the inequalities
(72)$$\frac{\partial S}{\partial a_{k}}\delta a_{k}\leq 0,\;k=1,2,\ldots$$
Here δa is the cone of variations of the vector *a* that are admissible with respect to the inclusion $a\in V_{a}$.

Please note that the phase trajectory corresponding to an optimal cyclic process has no self-intersections [13].

## 5. Estimation of the Efficiency of Transition to a Cyclic Process

### 5.1. Conditions of Equivalence and Efficiency of a Cyclic Extension

The optimal cyclic mode problem (Equation 63)–(Equation 66) (we refer to it as Problem C) is an extension of a non-linear programming problem. Indeed, imposing the additional constraints x=const and u=const on the solution of this problem, we obtain the following optimal static mode problem (Problem S):(73)$$I_{S}=f_{0}\left(x,u,a\right)\to \max/\begin{array}{c} f_{\nu }\left(x,u,a\right)=0,\;\varphi_{j}\left(x,u,a\right)=0\\ u\in V_{u},\;a\in V_{a},\;\nu =\bar{1,m},\;j=1,2. \end{array}$$
Since the set of admissible solutions of problem (Equation 63)–(Equation 66) is larger than that of Problem S, it follows that
(74)$$I_{S}^{*}\leq I_{C}^{*}.$$
where $I_{C}^{*}$ denotes the value of the optimal cyclic mode problem.

One of the problems in designing cyclic processes consists of distinguishing a class of problems for which inequality (Equation 74) turns into an equality, i.e., the cyclic extension is equivalent to the static problem. An important role in solving this problem is played by the Lagrangian function of Problem S,
(75)$$R_{S}=f_{0}\left(x,u,a\right)+\sum_{\nu }\lambda_{\nu }f_{\nu }\left(x,u,a\right)+\sum_{j}\xi_{j}\varphi_{j}\left(x,u,a\right)$$
To determine whether a cyclic process is equivalent to a static one or efficient without solving problem (Equation 63)–(Equation 66), we form averaged problems, which are in turn extensions for Problem S or C or for both. Comparing the values of these problems with $I_{C}^{*}$, we find conditions for the equivalence of a cyclic extension.

*An upper bound for $I_{C}^{*}$ and sufficient conditions for the equivalence of a cyclic extension*. Let us enlarge the set of admissible solutions of Problem C by removing the differential equations (Equation 63). We obtain Problem $\bar{S}$, which we call an *estimating* problem:
(76)$$I_{\bar{S}}={\bar{f_{0}\left(x,u,a\right)}}^{x,u}\to \max/\begin{array}{c} {\bar{f_{\nu }\left(x,u,a\right)}}^{x,u}=0,\;{\bar{\varphi_{j}\left(x,u,a\right)}}^{x,u}=0\\ \nu =\bar{1,m},\;u\in V_{u},\;a\in V_{a},\;j=\bar{1,r}. \end{array}$$
Clearly,
(77)$$I_{\bar{S}}^{*}\geq I_{C}^{*},$$
and Problem $\bar{S}$ is an averaged extension of Problem S with the variables *x* and *u* and the parameters *a*. The roles of the variables *x* and *u* in the conditions of Problem $\bar{S}$ are similar, and we unite these variables and denote them by y=(x,u). In shorthand notation, this problem has the form
(78)$$I_{\bar{S}}={\bar{f_{0}\left(y,a\right)}}^{y}\to \max/{\bar{f_{\nu }\left(y,a\right)}}^{y}=0,\;{\bar{\varphi_{j}\left(y,a\right)}}^{y}=0,\;\nu =\bar{1,m},\;j=\bar{1,r}.$$
The value of problem (Equation 78) as an extension of the optimal static mode problem can be expressed in terms of the function $R_{S}$ as
(79)$$I_{\bar{S}}^{*}=\underset{\lambda ,\xi }{\inf}\underset{y}{\sup}R_{S}\left(y,a^{*},\lambda ,\xi \right).$$
For determining the vector of parameters, we have the condition
(80)$${\left[\frac{\partial }{\partial a}{\bar{R_{S}\left(y,a,\lambda ,\xi \right)}}^{y}\right]}_{a=a^{*}}=0.$$
If $a^{*}$ lies inside $V_{a}$, then condition (Equation 80) reduces to the condition of stationarity of $R_{S}$ with respect to *a*.If the value $I_{5}^{*}$ given by (Equation 79) equals $I_{S}^{*}$ (i.e., Problem $\bar{S}$ has a unique base solution), then inequalities (Equation 74) and (Equation 77) imply $I_{C}^{*}=I_{S}^{*};$ i.e., the static mode cannot be improved by passing to a cyclic mode. If $I_{\vec{S}}^{*}>I_{S}^{*}$, then the difference $\Delta_{\bar{S}}$ between these values gives an upper bound for the possible gain from the passage to a cyclic mode.*A lower bound for $I_{C}^{*}.$ Quasi-static and sliding modes*. Consider the case when x(t) and u(t) vary so that the time derivatives of x(t) can be neglected. Then the relations between *x* and *u* are given, as in the static case, by f(x(t),u(t),a)=0 for all t. The corresponding modes are said to be quasi-static. The problem of an optimal choice of x(t) and u(t) under the quasi-static conditions (Problem QS) has the form
$$I_{\mathrm{QS}}=\frac{1}{\tau }\int_{0}^{\tau }f_{0}\left(x,u,a\right)dt\to \max/f\left(x,u,a\right)=0,\;\int_{0}^{\tau }\varphi \left(x,u,a\right)dt=0,\;u\in V_{u},\;a\in V_{a}.$$
or, in shorthand notation,
(81)$$I_{\mathrm{QS}}={\bar{f_{0}\left(y,a\right)}}^{y}\to \max/{\bar{\varphi (y,a)}}^{y}=0,\;y\in V_{y},\;a\in V_{a}.$$
Here y=(x,u), and the set $V_{y}$ is determined by the conditions $u\in V_{u},a\in V_{a},$ and f(x,u,a)=0.Since any solution of Problem QS is admissible for Problem C, it follows that
(82)$$I_{\mathrm{QS}}^{*}\leq I_{C}^{*}.$$
At the same time, the value $I_{QS}^{*}$ of Problem QS, being the value of an averaged problem, is given by the expression
(83)$$I_{\mathrm{QS}}^{*}=\underset{\xi }{\inf}\left\{\underset{y}{\sup}\left[f_{0}\left(y,a^{*}\right)+\xi \varphi \left(y,a^{*}\right)\right]/f\left(y,a^{*}\right)=0,u\in V_{u}\right\}.$$
Here $a^{*}$ is the optimal value of *a* subject to the constraint
(84)$$\frac{\partial }{\partial a}{\left[{\bar{f_{0}\left(y,a\right)}}^{y}+\left(\xi ,{\bar{\varphi (y,a)}}^{y}\right)+\sum_{i=0}^{r}\lambda^{i}f\left(y^{i},a\right)\right]}_{a^{*}}\delta a\leq 0.$$
in which δa is the set of variations allowed by the inclusion $a\in V_{a}$.We choose the Lagrange multipliers $\lambda^{i}$ in (Equation 84) so that $f(y^{i},a)=0$ for any base value $y^{i}$ of the vector *y*. The number of base values of *y* is determined by the dimension *r* of the vector function φ; thus, the problem takes the form
(85)$${\bar{f}}_{0}=\sum_{i=0}^{r}\gamma_{i}f_{0}\left(y^{i},a\right),\;\bar{\varphi }=\sum_{i=0}^{r}\gamma_{i}\varphi \left(y^{i},a\right),\;\sum_{i=0}^{r}\gamma_{i}=1,\;\gamma_{i}\geq 0.$$
Consider the case when the control vector in the steady state of the system changes with a frequency so high that the state vector *x* remains virtually constant. Such a mode is called a *sliding* steady mode. The optimization problem for such a mode is formulated as
(86)$$I_{\mathrm{SL}}=\bar{f_{0}{\left(b,u\right)}^{u}}\to \sup/{\bar{f(b,u)}}^{u}=0,\;u\in V_{u},\;{\bar{\varphi (b,u)}}^{u}=0,\;b\in V_{b}.$$
This problem is known as Problem SL. In (Equation 86), *b* denotes the vector formed by *x* and *a*. This mode is the limit case of the cyclic mode, so we have
$$I_{\mathrm{SL}}^{*}\leq I_{C}^{*}.$$
Problem (Equation 86) is an averaged extension of Problem S with two types of variables; its value is given by
(87)$$I_{\mathrm{SL}}^{*}=\min_{\lambda ,\xi }\max_{u}R\left(u,b^{*},\lambda ,\xi \right)=\min_{\lambda ,\xi }\max_{u\in V_{u}}\left[f_{0}\left(u,b^{*}\right)+\lambda f\left(u,b^{*}\right)+\xi \varphi \left(u,b^{*}\right)\right],$$
where $b^{*}$ satisfies the condition
(88)$$\frac{\partial }{\partial b}{\bar{R\left(u,b^{*},\lambda ,\xi \right)}}^{u}\delta b\leq 0.$$
The number of base values of the vector function *u* in Problem SL is at most m+r+1.A necessary condition for the efficiency of the transition to a cyclic mode can be stated in terms of $I_{\mathrm{QS}}^{*}$ and $I_{\mathrm{SL}}^{*}$. Consider the quantity
$$I_{K}=\max\left[I_{\mathrm{QS}}^{*},I_{\mathrm{SL}}^{*}\right].$$If $I_{K}$ is greater than $I_{S}^{*}$, then the passage to a cyclic mode is efficient, and the difference
$$\Delta_{K}=I_{K}-I_{S}^{*}$$
provides a lower bound for the efficiency.

### 5.2. The Frequency Criterion for the Efficiency of the Passage to a Cyclic Mode

Suppose that an optimal static mode $x^{0}$, $u^{0}$ in Problem S is known. As above, it is required to determine whether the cyclic extension of Problem S is efficient. In [14], a frequency criterion for the efficiency of a cyclic mode was proposed. This criterion is based on the analysis of the increment in the optimality criterion *I* as compared to its maximal static value $I^{0}$ for small harmonic oscillations of the control about $u^{0}$.

Let $\lambda^{0}$ and $\mu^{0}$ be the values of Lagrange multipliers λ and μ corresponding to the optimal static mode in the Lagrangian function
$$R=f_{0}\left(x,u\right)+\sum_{i}\lambda_{i}f_{i}\left(x,u\right)+\sum_{j}\mu_{j}\varphi_{j}\left(x,u\right)$$
for Problem S.

In a neighborhood of the optimal static mode and the corresponding Lagrange multipliers, we calculate the first and second derivatives of the functions that determine the problem with respect to *x* and *u* (if *x* and *u* are vectors, then these derivatives are matrices):$$A=\frac{\partial f}{\partial x},\;B=\frac{\partial f}{\partial u},\;P=\frac{\partial^{2}R}{\partial x^{2}},\;Q=\frac{\partial^{2}R}{\partial x\partial u},\;H=\frac{\partial^{2}R}{\partial u^{2}},\;K=\frac{\partial \varphi }{\partial x},\;M=\frac{\partial \varphi }{\partial u}.$$
In a neighbourhood of the optimal static mode, the increment of the functional *I* under small variations δx(t) and δu(t) is given by
$$\Delta I=\frac{1}{2T}\int_{0}^{T}\left(\delta x^{\prime }P\delta x+\delta x^{\prime }Q\delta u+\delta u^{\prime }Q^{\prime }\delta x+\delta u^{\prime }H\delta u\right)dt.$$
The transition to a cyclic mode is efficient if there is a variation δu such that the quantity ΔI is positive under the linearized constraints (Equation 63), i.e.,
(89)$$\delta \dot{x}=A\delta x+B\delta u,\;\delta x\left(T\right)=\delta x\left(0\right).$$
To get rid of these constraints, we consider only harmonic variations, i.e., those of the form
$$\delta u\left(t\right)=\sum_{\nu =-\infty }^{\infty }u_{\nu }e^{i\nu \frac{2\pi }{T}t}.$$
Applying the Fourier transform to the linear differential constraints (Equation 89), we obtain
$$\delta x\left(i\omega \right)=\delta u\left(i\omega \right)\frac{B}{i\omega E-A}=\delta u\left(i\omega \right)W\left(i\omega \right).$$
Here *E* is the identity matrix (E=1 for scalar *x*). It is assumed that the matrix *A* has no eigenvalues with zero real part; otherwise, small deviations δu(t) may correspond to large deviations δx(t), and the linearization may be incorrect.

Let us express the quantity ΔI by Parseval’s identity in the frequency domain, replacing δx(iω) by its expression in terms of δu. The increment in the criterion under harmonic oscillations of the control with frequencies that are multiples of 2π/T takes the form
$$\Delta I=\frac{1}{2}\int_{-\infty }^{\infty }\delta u^{\prime }\left(-i\omega \right)A\left(\omega \right)\delta u\left(i\omega \right)d\omega .$$
Here A(ω) is defined by the matrices *P*, *Q*, and *H* and the relation between δu and δx; it is easy to show that
$$A\left(\omega \right)=W^{\prime }\left(-i\omega \right)PW\left(i\omega \right)+Q^{\prime }W\left(i\omega \right)+W^{\prime }\left(-i\omega \right)Q+H,$$
where the prime denotes transposition.

For the scalar problem, we have
A(ω)=P|W(iω)|2+2QReW(iω)+H
If the matrix A(ω) for some ω is such that the integrand in the expression for ΔI is positive for at least one vector δu, then the static mode can be improved and the passage to a cyclic mode is efficient.

For the scalar problem, we have
$$\Delta I=\frac{1}{2}\int_{-\infty }^{\infty }{\left|\delta u\left(i\omega \right)\right|}^{2}A\left(\omega \right)d\omega ,$$
and the static mode improves if A(ω) is positive for some ω.

### 5.3. Lyapunov Problems

For an important class of problems, the inequality (Equation 77) turns into an equality. In these problems, the functions $f_{0}$, *f*, and φ in relations (Equation 63)–(Equation 66) depend only on *u* and *a*, so that
(90)$$\dot{x}=f\left(u,a\right)$$
Such equations are called *Lyapunov-type equations*, and the corresponding problems are known as Lyapunov problems. If we discard equations (Equation 63), which have the form (Equation 90), in Problem C, thereby transitioning to Problem S, then we can find its solution $u^{*}\left(t\right)$, $a^{*}$. Substituting this solution into equation (Equation 90), we determine an optimal trajectory. Clearly, in this problem, $I_{\bar{S}}^{*}=I_{C}^{*}$, $u^{*}\left(t\right)$ takes at most m+r+1 base values, and the function $x^{*}\left(t\right)$ is a polygonal line with at most m+r (internal) vertices.

Problems that include, in addition to Lyapunov-type equations, equations of the form
$${\dot{x}}_{\nu }=f_{\nu }\left(u,a\right)F_{\nu }\left(x_{\nu }\right)$$
can also be reduced to Lyapunov problems. Indeed, such equations can be reduced to the form (Equation 90) by the change
(91)$$y_{\nu }\left(x_{\nu }\right)=\int \frac{dx_{\nu }}{F_{\nu }\left(x_{\nu }\right)},$$
so that ${\dot{y}}_{\nu }=f_{\nu }\left(u,a\right).$ The optimal solution $y_{\nu }\left(t\right)$ is piecewise linear, and $x_{\nu }\left(t\right)$ can be found from (Equation 91) by solving the equation
$$\frac{dx_{\nu }}{F_{\nu }\left(x_{\nu }\right)}=y_{\nu }\left(t\right)dt.$$

## 6. Average Optimization in Finite-Time Thermodynamics

The field of finite-time thermodynamics is one of the most important examples of application of averaged optimization techniques. The reasons for this are the following:Problems of optimal thermodynamic cycles.There are a very important kind of thermodynamic systems — intermediary ones. These systems contact different subsystems (reservoirs) alternately while producing power and thus lowering the irreversibility arising from a continuous contact of the above-mentioned subsystems. The main example here is the heat engine, where the working fluid contacts two sources of different temperature.One of the most essential problems in finite-time thermodynamics is the problem of maximum average power of heat engines, when the average rate of the heat flow from the hot source is given.Similar problems arise also in absorption-desorption systems, where the working fluid contacts with the multi-component mixture and picks one component out from one source, releasing it at another one.In reverse cycles, the working fluid obtains the energy from the exterior system. Upon contact with the source that loses energy or matter in the regular cycle, the working fluid enriches it with the corresponding resource.In all of these problems, the working fluid restores its state at the beginning of every cycle. One needs to average all of the variables determining the process.Relations between intensive and extensive variables are Lyapunov-type equations. Thermodynamic variables are divided into two classes: intensive (temperature, pressure, chemical potential, …) and extensive (volume, internal energy, entropy, amount of substance, …) ones. Flow rates of transport processes between subsystems depend only on intensive variables. This value determines in turn the rate of change of extensive variables. This means that equations determining the change of state of the thermodynamic system have the form [10,11,12]:
(92)$$\frac{dZ_{j}}{dt}=F_{j}\left(u_{i},u_{j}\right).$$
Here *i* and *j* are indices of the contacting subsystems, *u* is the vector of intensive variables, *Z* is the vector of extensive variables. Equations of this type are called Lyapunov-type equations earlier in this paper. The right hand side of these equations does not depend on *Z*, and the increase of *Z* is determined by the average value of the function *F*. As we have shown above, one can obtain the limiting capabilities of systems characterized by Lyapunov-type equations using techniques of the averaged optimization.

## 7. Example: Averaged Optimization of a Heat Engine

### 7.1. Maximum Average Power Output

We will assume that there is a heat engine with constant temperature of sources $T_{+}$ and $T_{-}$ [15]. If we denote the temperature of the source contacting with the working fluid at the moment as $T_{n}$ and the temperature of the working fluid itself as *T*, we will obtain that the average output per cycle is
(93)$$\bar{p}=\bar{q(T_{n},T)}.$$
Now we can formulate the averaged optimization problem, given that the average entropy generation within the working fluid per cycle is zero:(94)$$\bar{q(T_{n},T)}\to \max_{T},/\begin{array}{c} \bar{\left(\frac{q(T_{n},T)}{T}\right)}=0,\\ T_{n}=\left(T_{+};T_{-}\right),\;T>0. \end{array}$$
This is the problem in the form (Equation 6). Using the algorithm described earlier (see (Equation 42)–(Equation 44)) we find the number of base points is two. This means that the Lagrange function
(95)$$R=q\left(T_{n},T\right)+\lambda \frac{q(T_{n},T)}{T}=q\left(T_{n},T\right)\left(1+\frac{\lambda }{T}\right)$$
has two maxima, so
(96)$$\begin{array}{c} T_{1}=\underset{T}{\arg\max}\;q\left(T_{+},T\right)\left(1+\frac{\lambda }{T}\right),\\ T_{2}=\underset{T}{\arg\max}\;q\left(T_{-},T\right)\left(1+\frac{\lambda }{T}\right). \end{array}$$
Both maxima are global and therefore they must be equal [16]. It means that the Lagrange multiplier is the solution of
(97)$$q\left(T_{+},T_{1}\right)\left(1+\frac{\lambda }{T_{1}}\right)=q\left(T_{+},T_{2}\right)\left(1+\frac{\lambda }{T_{2}}\right).$$
When the heat transfer law is linear
(98)$$q\left(T_{+},T\right)=\alpha_{+}\left(T_{+}-T\right),\;q\left(T_{-},T\right)=\alpha_{-}\left(T_{-}-T\right),$$
solution of equations (Equation 94)–(Equation 96) with notation $\alpha =\frac{\alpha_{+}\alpha_{-}}{{\left(\sqrt{\alpha_{+}}+\sqrt{\alpha_{-}}\right)}^{2}}$ leads to
(99)$$p_{max}=\alpha {\left(\sqrt{T_{+}}-\sqrt{T_{-}}\right)}^{2}.$$

The relationship between entropy generation and heat flows is shown at Figure 3. It is clear from this picture that the point of maximum power output lies on the convex hull of original output curves, so it is attainable only when averaged control is used.

### 7.2. Maximum Efficiency

When the power output $p_{0}$ is given, the problem of maximum efficiency is equivalent to the problem of minimum entropy generation within the system. Using again that the average entropy generation within the working fluid is zero for a cyclic process, we obtain the problem:(100)$$-\sigma =\bar{\left(\frac{q(T_{n},T)}{T_{n}}\right)}\to \max_{T},/\begin{array}{c} \bar{q(T_{n},T)}=p_{0},\\ \bar{\left(\frac{q(T_{n},T)}{T}\right)}=0,\\ T_{n}=\left(T_{+};T_{-}\right),\;T>0. \end{array}$$
One may notice that this problem allows three base points in general, because there are three averaging operations in (Equation 100). This is the case when the entropy generation as a function of $q(T_{+},T)$ is not convex. We will not consider this case here, because this function is convex for most of heat transfer laws.

Another possibility corresponds to two base points. In this case, we have the following equations for $T_{1}$ and $T_{2}$:(101)$$\begin{array}{c} T_{1}=\underset{T}{\arg\max}\;\left[q\left(T_{+},T\right)\left(\frac{1}{T_{+}}+\lambda +\frac{\mu }{T}\right)-\lambda p_{0}\right],\\ T_{2}=\underset{T}{\arg\max}\;\left[q\left(T_{-},T\right)\left(\frac{1}{T_{-}}+\lambda +\frac{\mu }{T}\right)-\lambda p_{0}\right]. \end{array}$$
These maxima must be equal, which leads to:(102)$$q\left(T_{+},T\right)\left(\frac{1}{T_{+}}+\lambda +\frac{\mu }{T}\right)=q\left(T_{-},T\right)\left(\frac{1}{T_{-}}+\lambda +\frac{\mu }{T}\right).$$
The averaged constraints must also be satisfied:(103)$$\begin{array}{c} \gamma q\left(T_{+},T_{1}\right)+\left(1-\gamma \right)q\left(T_{-},T_{2}\right)=p_{0},\\ \gamma \frac{q(T_{+},T_{1})}{T_{1}}+\left(1-\gamma \right)\frac{q(T_{-},T_{2})}{T_{2}}=0. \end{array}$$
Equations (Equation 101)–(Equation 103) allow one to find values of $T_{1}$, $T_{2}$, λ, μ and γ.

For the linear heat transfer law we have the following value of maximum efficiency:(104)$$\eta_{max}\left(p\right)=\frac{1}{2}\left(\frac{p}{\alpha T_{+}}+\eta_{c}\right)\pm \sqrt{\frac{1}{4}{\left(\frac{p}{\alpha T_{+}}+\eta_{c}\right)}^{2}-\frac{p}{\alpha T_{+}}}.$$
When p→0, the value of (Equation 104) approaches $\eta_{c}$ (Carnot efficiency) and when $p=p_{max}$ (Equation 99), we have
(105)$$\eta_{max}\left(p_{max}\right)=1-\sqrt{\frac{T_{-}}{T_{+}}}=1-\sqrt{1-\eta_{c}},$$
which is the well-known result of Novikov [17], Chambadal [18], Curzon and Ahlborn [19]. Important results for other types of heat transfer laws and different processes are presented in [20,21,22].

## 8. Results

We obtained the general necessary conditions of optimality for averaged optimization problems. These conditions can be written down using the algorithmic procedure given in the paper, which allows one to use them for problems of any structure. We showed how these techniques can be applied to the problems of finite-time thermodynamics leading to new results in the field.

## Figures and Tables

**Figure 1 entropy-22-00912-f001:**
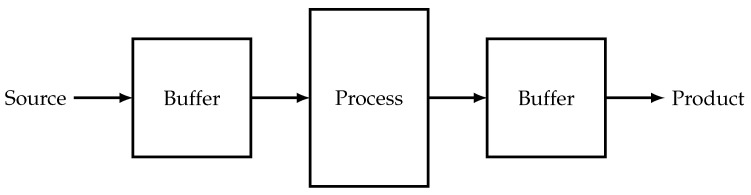
Flowsheet of a simple process with averaging of both source and product flows.

**Figure 2 entropy-22-00912-f002:**
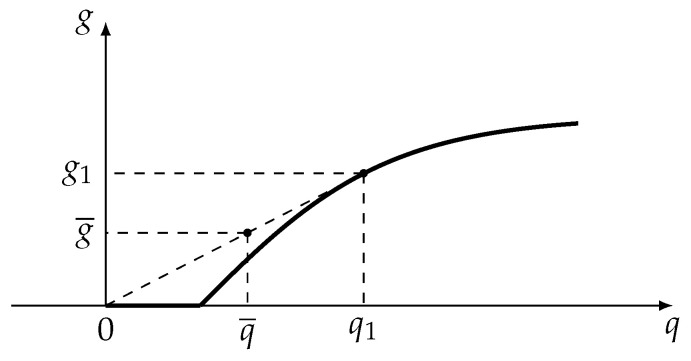
Relationship between production and consumption. Effect of averaging.

**Figure 3 entropy-22-00912-f003:**
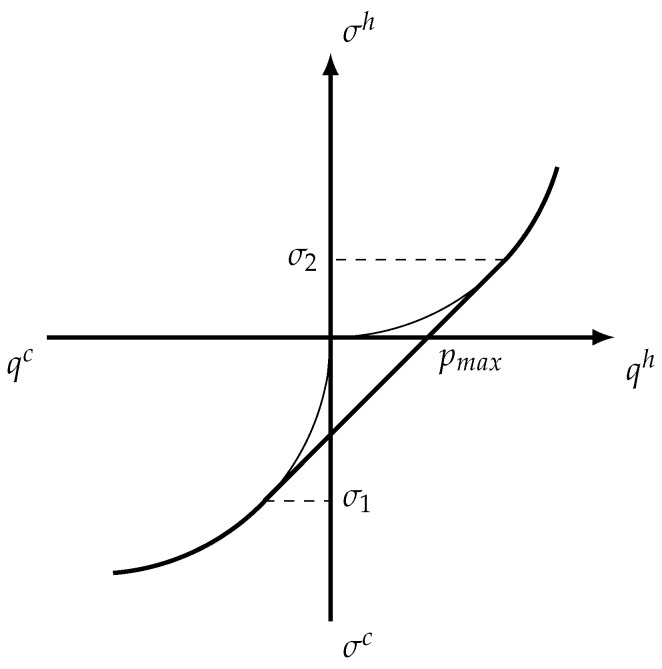
Relationship between entropy generation and heat flows and its convex hull. Here $q^{h}$ and $q^{c}$ are heat exchange rates upon contact with the hot and cold reservoirs, respectively, and $\sigma^{h}$, $\sigma^{c}$ are the corresponding entropy generation rates. The optimal solution is attained when $\sigma^{c}=\sigma_{1}$, $\sigma^{h}=\sigma_{2}$, $q^{1}=q^{c}\left(\sigma_{1}\right)$, $q^{2}=q^{h}\left(\sigma^{2}\right)$ and $p_{max}=q^{1}+q^{2}$.

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
