# Peer review of "Averaged Optimization and Finite-Time Thermodynamics"

_entropy, 2020, doi:10.3390/e22090912_

Round 1
Reviewer 1 Report
The paper is a beautiful introduction to the theorem of average optimization. I have never seen a clearer of more thorough treatment.
Author Response
Thank you for your review. We are glad that you liked our paper and hope that it will be of interest to the readers of Entropy.
Reviewer 2 Report
The paper examines nonlinear optimization problems involving averaging of the control variables, the objective functionals, and the constraint functionals.
Various sub-classes of these optimization problems are introduced. It is shown how these can be algorithmically recast in canonical form such that general optimality conditions can be derived. The special relevance of averaged optimization problems to the field of finite-time
thermodynamics is highlighted. Real-life examples of such optimization problems are provided throughout the paper. In particular, the case of averaged optimization of a heat engine is discussed in detail, thus showing the relevance and efficacy of the proposed framework.
I found this paper very interesting and insightful. The treatment is quite abstract and general, while clearly showing the importance of this approach for optimization problems arising in finite-time thermodynamics. The paper is well structured, and the methods and results are carefully described. In my opinion, this work will be of great interest to the readers of Entropy, and therefore I recommend it for publication.
As a general suggestion, I believe the paper could be improved by providing a more pedagogical introduction to the various mathematical concepts necessary to follow the study. In particular, I think it would be helpful to briefly introduce (perhaps in the appendix) the following notions: convex hull of a function, Carathéodory’s theorem, Kuhn–Tucker theorem.
Moreover, some of the results could probably be made easier to follow if accompanied by illustrative figures presenting low-dimensional problems that can be visualized in three-dimensional space by using e.g. graphs of the objective functions. So I suggest adding one or two figures to the first part of the paper, if possible.
Finally, I noticed a few minor typos:
-In the first paragraph of page 5: the zero in the expression f*0(C) should probably be a subscript.
-In the first line after line 119: "A relationship between problem (6) and the Lagrangian function of the NLP problem". It is not clear if this is a title. Otherwise the sentence misses a verb.
-Second line of equation 12: there is the word "quad" instead of space.
Author Response
Thank you for your careful review. We have fixed typos you mentioned and extended our paper with key definitions. Also, we have added a new figure representing the problem and its solution for the two-dimensional case. All of the changes made are marked red in the new version of the manuscript.